# Zero-shot Inversion Process for Image Attribute Editing with Diffusion Models

## Abstract

Denoising diffusion models have shown outstanding performance in image editing. Existing works tend to use either image-guided methods, which provide a visual reference but lack control over semantic coherence, or text-guided methods, which ensure faithfulness to text guidance but lack visual quality. To address the problem, we propose the Zero-shot Inversion Process (ZIP), a framework that injects a fusion of generated visual reference and text guidance into the semantic latent space of a *frozen* pre-trained diffusion model. Only using a tiny neural network, the proposed ZIP produces diverse content and attributes under the intuitive control of the text prompt. Moreover, ZIP shows remarkable robustness for both in-domain and out-of-domain attribute manipulation on real images. We perform detailed experiments on various benchmark datasets. Compared to state-of-the-art methods, ZIP produces images of equivalent quality while providing a realistic editing effect.

## 1 Introduction

Manipulating real-world images with natural language has long been a challenge in image processing. Recently, denoising diffusion models (DDMs) have shown substantial success in text-to-image tasks, such as Imagen (Saharia et al., 2022), Dall-E (Ramesh et al., 2021), and Stable Diffusion (Rombach et al., 2022). These text-to-image models produce diverse, highly coherent, and realistic images that align well with text prompts. However, attribute manipulation on real images is still a challenging problem. In this paper, we aim to utilize these novel foundational models to manipulate real images in a controllable and semantically coherent manner.

Figure 1 briefly illustrate currently popular methodologies. On the one hand, much effort has been put into text-guided image editing. Along with the development of the Natural Language Processing (NLP) technique, e.g., Generative Pre-Training (GPT) (OpenAI, 2023; Brown et al., 2020), many previous works (Bao et al., 2023; Hertz et al., 2022) develop image-editing teniques with the guidance of textual prompts. However, the importance of the visual reference is ignored in these methods. Though text-guided methods maintain faithfulness to the target semantic, it is difficult to learn fine-grained visual patterns from textual features in the absence of a visual prior. Using textual semantics alone lacks visual reference, resulting in sketchy semantic manipulation. Especially if the desired semantic is out of the domain, text-guided editing fails.

On the other hand, image-guided methods attract a large amount of attention. Image-guided editing can easily make style transfer (Choi et al., 2021) and item replacement (Jia et al., 2023). With visual reference, generators can directly insert ready-made visual patterns into images. However, image-guided approaches lack intuitive control over semantic coherence, and it is ambiguous to specify which attribute to refer from the reference image.

In recent times, there has been a growing interest in the field of real image editing. Prompt2Prompt (Hertz et al., 2022) has shed light on the potential of cross-attention layers for semantic editing in image generation. Following this, InstructPix2Pix (Brooks et al., 2023) and Null-text Inversion (Mokady et al., 2022) employed the same principle for semantic editing of real images. However, one limitation of these approaches is their inability to precisely control specific attributes during the editing process. For instance, as demonstrated in Figure 1(a), the glasses added by Null-text Inversion lacks a predefined style, making it challenging to confirm the exact appearance of the added glasses in advance. The visual prompt can serve to accurately delineate the desired attributes. Jia et al. (2023) embeds specified items into the target image by encoding a reference

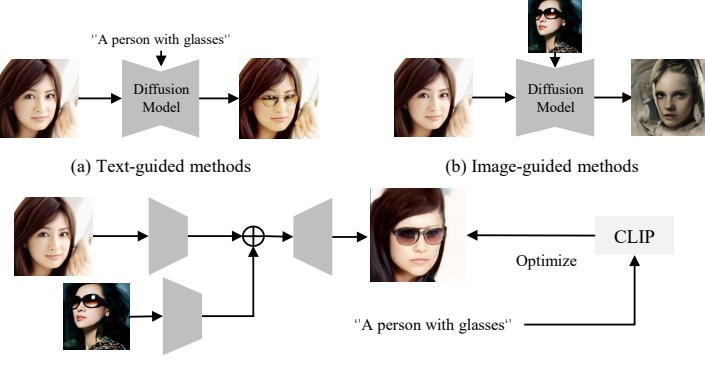

Figure 1: **Editing methods for denoising diffusion models.** (a) Text-guided methods lacks a predefined style of the attribute, where the glasses are generated by models. (b) Image-guided methods suffer ambiguity and distorted results. (c) Our method achieves specific, controllable and high-quality manipulation, where the style of glasses is in accordance with the reference image.

image. VISII (Nguyen et al., 2023) amalgamates both textual and visual prompts to learn a style transfer from example pairs, representing the "before" and "after" images of an edit. Nevertheless, as shown in Figure 1(b), when a reference image is employed for attribute editing of real images, the final rendition is effected by the reference image, which causes image distortion.

In this paper, we propose Zero-shot Inversion Process (ZIP) that injects a fusion of visual reference and text guidance into the semantic latent space of a *frozen* pre-trained diffusion model. As illustrated in Figure 1(c), our method takes advantage of the text guidance to provide intuitive control over the semantic coherence. Meanwhile, our methods refine the alignment of the text feature and the semantic latent space of the diffusion model by incorporating a visual reference. The incorporated reference image can be generated with a foundational text-to-image model. Thus, our method is zero-shot and avoids the bias of manual selection. To the best of our knowledge, it is the first attempt to integrate text guidance and image guidance in zero-shot image editing. Our method only needs to train an attribute encoder, a tiny neural network, without fine-tuning the pre-trained model. For both in-domain and out-of-domain attribute editing tasks, our method preserves faithfulness to text guidance while maintaining visual quality. Extensive experiments demonstrate that our method is generally applicable to various benchmark datasets (CelebA-HQ, LSUN-church, and LSUN-bedroom).

## 2 RELATED WORK

**Text-to-Image Synthesis:** Since the success of large language models, text prompts are widely used in image editing. Benefiting from the semantic information of text, Stabe Diffusion (Rombach et al., 2022) makes a powerful and flexible generator with the condition of the text. Diffusionclip (Kim et al., 2022) shows that a textual prompt allows DDMs to edit the images in a semantic latent space. Asyrp (Kwon et al., 2022) reveals that diffusion models already have a semantic latent space. Imagen Editor (Wang et al., 2022) uses the masks as input to point out the area of edit in the image, which can gather up the semantic information into the target. Though these works show a significant process in the editing of some attributes, only by text prompt, they just can generate ordinary attributes such as colors or common shapes in the image. To generate new visual features, such as glasses and other decorations in the images, the visual information should be taken into account.

**Image-to-Image Synthesis:** In contrast, image-guided approaches use the reference image as a condition to generate corresponding images. On the one hand, the features from the reference image are used to adjust the target image. There are many typical tasks such as style transfer (Choi et al., 2021; Meng et al., 2021) and inpainting (Lugmayr et al., 2022), where the reference image is viewed as an auxiliary feature. SDEdit (Meng et al., 2021) uses the stroke-based images to generate faithful images with the original images. On the other hand, the reference image is directly inserted into the target image. Jia et al. (2023) makes remarkable performance by using the reference image for personalized syntheses, such as the replacement of the items. Mystyle (Nitzan et al., 2022) adopts a pre-trained StyleGAN for personalized face generation by using the images of the generic face prior.

However, due to the ambiguity of the attribute choice in one image, image-guided approaches often make distorted images or incorrect manipulation. By using the text prompt, our ZIP circumvents this drawback with the alignment of visual features and semantic information from text.

**Attribute Editing**  Many previous works (Kwon et al., 2022; Wallace et al., 2022; Daras and Dimakis, 2022) have focused on image editing based on large-scale generative models. On the one hand, these models mainly focus on editing DDM-generated images rather than real images. By adding, replacing, or modifying corresponding features, Prompt2prompt (Hertz et al., 2022) can change the items of a DDM-generated image. However, it is difficult to learn textual features corresponding to the real image. On the other hand, some models can only achieve sketchy editing for real images. For example, the background replacement is made in Direct Inversion (Elarabawy et al., 2022). The replacement of items in the real images is achieved in Jia et al. (2023). Some in-domain attributes, such as the age, gender, and expression of humans, are modified by Asyrp (Kwon et al., 2022). However, these models are incomplete for the attributes which need both visual and semantic information such as wearing glasses.

## 3 PRELIMINARY

In this section, we provide a concise overview of the foundational knowledge of the diffusion model and the Contrastive Language-Image Pre-Training (CLIP) model. The diffusion model is employed for the generation of visual prompts, facilitating the editing of real images. Subsequently, CLIP is utilized to adjust the images in accordance with the textual prompt.

### 3.1 DIFFUSION MODEL

Denoising diffusion models (DDMs) produce more realistic samples than deep generative models before, such as GANs (Karras et al., 2020). A typical DDM includes two stages: the forward process to add Gauss noise to the original data and the reverse process to denoise samples until an image. Denoising Diffusion Probabilistic Model (DDPM) (Ho et al., 2020), starting from white noise, progressively denoises it into an image. Denoising Diffusion Implicit Models (DDIM) (Song et al., 2020a) reduce the number of iterations by taking generated results of different stages into the condition of generation.

The forward process diffuses the original data $x_0$ with Gaussian noise, indexed by a real vector $\sigma \in \mathbb{R}^T_{\geq 0}$:

$$q_\sigma(x_{1:T}|x_0) := q_\sigma(x_T|x_0) \prod_{t=2}^T q_\sigma(x_{t-1}|x_t, x_0), \tag{1}$$

where $q_\sigma(x_T|x_0) = \mathcal{N}(\sqrt{\alpha_{T-1}}, (1 - \alpha_T)\boldsymbol{I})$. The corresponding reverse process is

$$x_{t-1} = \sqrt{\alpha_{t-1}/\alpha_t} \left(x_t - \sqrt{1 - \alpha_t}\epsilon_\theta(x_t, t)\right) + \sqrt{1 - \alpha_{t-1} - \sigma_t^2} \cdot \epsilon_\theta(x_t, t) + \sigma_t \epsilon_t, \tag{2}$$

where $\epsilon_t \sim \mathcal{N}(\boldsymbol{0}, \boldsymbol{I})$ is standard Gaussian noise, $\alpha_t$ is the parameter based on the forward process, $\sigma_t = \eta\sqrt{(1 - \alpha_{t-1})/(1 - \alpha_t)}\sqrt{1 - \alpha_t/\alpha_{t-1}}$, and $\epsilon_\theta(x_t, t)$ is a neural network to predict the noise in $x_t$. In this paper, we use a U-Net backbone which is introduced in the supplementary material. When $\eta = 1$ for all $t$, the process of Equation 2 becomes DDPM (Ho et al., 2020). As $\eta = 0$, it becomes DDIM and guarantees nearly perfect inversion (Song et al., 2020a).

### 3.2 CONTRASTIVE LANGUAGE-IMAGE PRE-TRAINING

Contrastive Language-Image Pre-Training (CLIP) (Radford et al., 2021) encapsulates generic and semantic information of image-text pairs. CLIP simultaneously learns an image encoder $E_I$ and a text encoder $E_T$ to indicate the similarity between images and texts. StyleGAN-NADA (Gal et al., 2022) produces images guided by a text prompt based on CLIP. Asyrp (Kwon et al., 2022) also uses the CLIP to fine tune the pre-trained diffusion model. Inspired by previous works, we adopt CLIP as guidance for the attribute modification:

$$\mathcal{L}_{\text{direction}}(i_{\text{out}}, t_{\text{target}}; i_{\text{edit}}, t_{\text{source}}) := 1 - \frac{\Delta I \cdot \Delta T}{\|\Delta I\|\|\Delta T\|}, \tag{3}$$

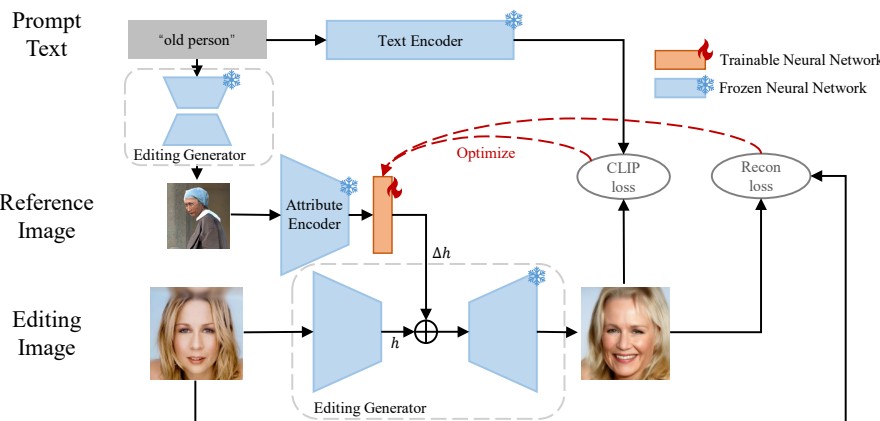

Figure 2: **The framework of ZIP.** A reference image, generated by the Visual Generator, is encoded into edited features denoted as $\Delta h$. These edited features are then integrated into the existing features $h$ of the editing image. The textual prompt contributes semantic information for the manipulation process. To streamline computation, the parameters of both the Attribute Encoder and Editing Generator are shared.

where $\Delta I = E_I(i_{\text{out}}) - E_I(i_{\text{edit}})$ and $\Delta T = E_T(t_{\text{target}}) - E_T(t_{\text{source}})$, for the generated image $i_{\text{out}}$, the editing image $i_{\text{edit}}$, the target prompt $t_{\text{target}}$, and the source prompt $t_{\text{source}}$.

## 4 METHOD

We present the Zero-shot Inversion Process (ZIP) as a method for attribute editing. The core framework of ZIP is outlined in Figure 2, where each constituent element of the framework is delineated. Subsequently, the principle of ZIP is elaborated in Section 4.1. Then, the optimization process and the associated loss function of ZIP are discussed in Section 4.2. Finally, we summary the real image editing via ZIP in Section 4.3.

Given an image $i_{\text{edit}} \in \mathbb{R}^{m \times n}$ and an attribute $t_{\text{attr}}$, our primary objective is to modify the input image $i_{\text{edit}}$ in accordance with the attribute $t_{\text{attr}}$. This endeavor results in the creation of an edited image, denoted as $i_{\text{out}}$. Initially, a textual prompt denoted as $t$ is formulated based on the specific attribute $t_{\text{attr}}$, an example being the prompt "an old person" associated with the "old" attribute. Subsequently, the Visual Generator produces a reference image $i_{\text{ref}}$ conditioned on the input $t$. In the third step, attribute features $\Delta h$ are extracted from $i_{\text{ref}}$ using the Attribute Encoder. Following this derivation, $\Delta h$ is integrated into the latent space of the Editing Generator. The resultant latent representation $h + \Delta h$ forms the basis for generating the target image $i_{\text{out}}$ through the Editing Generator.

As illustrated in Figure 2, our framework encompasses four primary components: Text Encoder, Visual Generator, Attribute Encoder, and Editing Generator. To obtain the visual attributes corresponding to the designated attribute, we utilize a text-image model as Visual Generator. Furthermore, when the attribute involves the addition of embellishments, such as the glasses, a reference image can be manually specified to ensure consistency of embellishments during editing. Both the textual prompt and the visual prompt, which are encoded by Text Encoder and Attribute Encoder respectively, are employed for editing in the Editing Generator. Detailed descriptions of these components are provided in Appendix B.

### 4.1 ZERO-SHOT INVERSION PROCESS

Figure 3 provides a thorough overview of the entire Zero-shot Inversion Process (ZIP). Figure 3(a) reveals its denoising process, where the feature $\Delta h$ from the reference image $i_{\text{ref}}$ is integrated into the original features $h$. Figure 3(b) illustrates the editing process in ZIP, where $i_{\text{edit}}$ is reverted to noise $x_T$ and subsequently restored to $\widetilde{x}_0$ as $i_{\text{out}}$ by ZIP.

As shown in Figure 3(a), the visual attributes extracted from the reference image $i_{\text{ref}}$ are integrated into the editing image $i_{\text{edit}}$ to amplify latent visual attributes that were hitherto unseen. For the generation of the reference image $i_{\text{ref}}$ under the influence of the prompt condition $t_{\text{target}}$, we employ

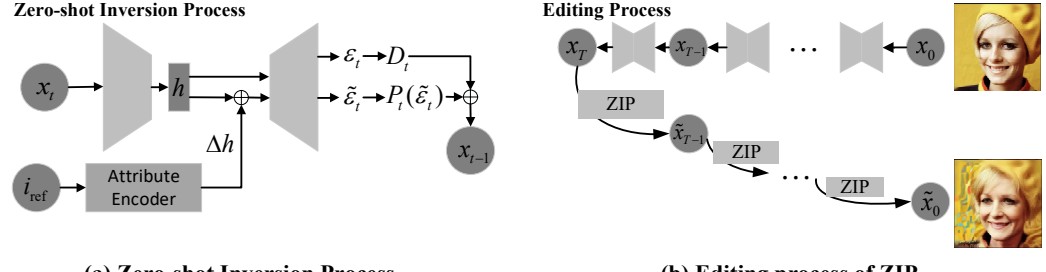

**(a) Zero-shot Inversion Process.**     **(b) Editing process of ZIP.**

Figure 3: **Zero-shot Inversion Process.** (a) The one step from $x_t$ to $x_{t-1}$ in Zero-shot Inversion Process. (b) The inversion process and the forward editing process for a real image.

UniDiffuser (Bao et al., 2023). Subsequently, through the utilization of the Attribute Encoder, we extract the corresponding features $\Delta h = E_A(i_{\text{ref}})$.

To facilitate latent manipulation on the images $x_0$ generated from $x_T$, a straightforward approach could involve directly modifying the Attribute Encoder to minimize the loss outlined in Equation 3. However, adopting this approach might engender distorted images or erroneous manipulations, as observed in prior works (Choi et al., 2021; Mokady et al., 2022).

An alternative approach entails the modification of the noise $\epsilon_t^\theta$ anticipated by the network during each sampling iteration. In brief, we can express the abridged version of Equation 2 as follows:

$$x_{t-1} = \sqrt{\alpha_{t-1}} \underbrace{\frac{1}{\sqrt{\alpha_t}} \left( x_t - \sqrt{1-\alpha_t}\epsilon_\theta(x_t,t) \right)}_{\text{predicted } x_0, P_t(\epsilon_\theta(x_t,t))} + \underbrace{\sqrt{1-\alpha_{t-1}-\sigma_t^2} \cdot \epsilon_\theta(x_t,t)}_{\text{direction to } x_t, D_t(\epsilon_\theta(x_t,t))} + \sigma_t\epsilon_t. \quad (4)$$

Nonetheless, a direct alteration of the noise $\epsilon_\theta$ in both $P_t$ and $D_t$ leads to mutual nullification, yielding an unchanged $p_\theta(x_{0:T})$. This phenomenon mirrors a form of destructive interference, as elucidated in Kwon et al. (2022, Theorem 1).

Hence, in order to circumvent the interference delineated in Equation 4, we resort to the utilization of an asymmetrical controllable reverse process within ZIP:

$$x_{t-1} = \sqrt{\alpha_{t-1}} P_t(\widetilde{\epsilon}_\theta(x_t,t)) + D_t(\epsilon_\theta(x_t,t)) + \sigma_t\epsilon_t. \quad (5)$$

Here, $\widetilde{\epsilon}_\theta(x_t,t)$ entails the adjustment of $\epsilon_\theta(x_t,t)$ grounded on the visual features $\Delta h$. This is achieved by introducing $\Delta h$ into the original feature maps $h_t$ derived from $x_t$.

## 4.2 OPTIMIZATION OF ZIP

Within the text-guided branch of ZIP, the text prompt $t$ is harnessed to facilitate the optimization of the ZIP generation process, relying on the CLIP model. Given the absence of ground truth labels in editing tasks, training the model follows a distinct approach compared to conventional vision tasks. As a result, we employ the CLIP loss to fine-tune the network. Aligning with the methodology presented in Avrahami et al. (2022), we employ the directional CLIP loss outlined in Equation 3 as our loss function:

$$\mathcal{L} = \lambda_{\text{clip}}\mathcal{L}_{\text{direction}}(\widetilde{P}_t, t_{\text{target}}; P_t, t_{\text{source}}) + \lambda_{\text{recon}}|x_{\text{out}}^t - x_{\text{edit}}^t|. \quad (6)$$

The modified $\widetilde{P}_t$ and the original $P_t$ correspond respectively to the formulations presented in Equation 5 and Equation 4. Here, $t_{\text{source}}$ and $t_{\text{target}}$ reference the text prompts outlined in Appendix B. The latter expression pertains to the reconstruction loss, which takes the form of the $L_1$ Loss between the generated image and the original image. This reconstruction loss effectively preserves the original features, thereby averting drastic alterations. To balance the aforementioned losses, the hyperparameters $\lambda_{\text{clip}}$ and $\lambda_{\text{recon}}$ are introduced.

During the training phase, the reference image $i_{\text{ref}}$ and the editing image $i_{\text{edit}}$ are subject to encoding by the Attribute Encoder and Editing Generator, in accordance with the architecture depicted in Figure 2. Consequently, these encoded representations manifest as $\Delta h$ and $h$ in the latent space, respectively.

The resulting output image $i_{\text{out}}$, generated by a *frozen* diffusion model (Editing Generator) with input $\Delta h + h$, is harnessed for computing the CLIP loss, thereby facilitating the training of the Attribute Encoder. In this process, updates are exclusively applied to the parameters of the Attribute Encoder, while the other components, including the Editing Generator, remain *frozen*.

### 4.3 IMAGE EDITING VIA ZIP

Given an image $i_{\text{edit}} \in \mathbb{R}^{m \times n}$ and an attribute $t_{\text{attr}}$, the visual features from reference image are integrated into the latent space of $i_{\text{edit}}$ in diffusion model as Equation 5, and the textual prompt is used to optimize the Attribute Encoder as Equation 6. We provide an illustration of our framework in Figure 2, and pseudocode in Algorithm 1.

---

**Algorithm 1** Zero-shot Inversion Process (ZIP)

---

**Input:** An editing image $i_{\text{edit}}$; A text prompt $t_{\text{attr}}$
$\qquad$ Editing Generator $\epsilon_\theta$; Visual Generator $G_V$; Attribute Encoder $E_A$ CLIP encoder $\xi_{clip}$
$\qquad$ Diffusion model timestep $T$; ZIP timestep $t_{\text{zip}}$
**Output:** A target image $i_{\text{out}}$
  1: Initialize $t_{\text{source}}$ and $t_{\text{target}}$ based on $t_{\text{attr}}$ $\qquad\qquad\qquad\qquad$ ▷ Get the textual prompt
  2: Generate the reference image $i_{\text{ref}} = G_V(t_{\text{target}})$ $\qquad\qquad\qquad$ ▷ Get the visual prompt
  3: Encode $\Delta h = E_A(i_{\text{ref}})$
  4: Get the noise image $x_0$ from $i_{\text{edit}}$ based on $\epsilon_\theta$
  5: **for** $i = 1, 2, \ldots, N$ **do**
  6: $\qquad$ **for** $t = T, T-1 \ldots, 0$ **do**
  7: $\qquad\qquad$ **if** $t > t_{\text{zip}}$ **then**
  8: $\qquad\qquad\qquad x_{t-1} = \sqrt{\alpha_{t-1}} P_t(\widetilde{\epsilon}_\theta(x_t, t)) + D_t(\epsilon_\theta(x_t, t)) + \sigma_t \epsilon_t.$ $\qquad$ ▷ Editing for $i_{\text{edit}}$
  9: $\qquad\qquad$ **else**
10: $\qquad\qquad\qquad x_{t-1} = \sqrt{\alpha_{t-1}} P_t(\epsilon_\theta(x_t, t)) + D_t(\epsilon_\theta(x_t, t)) + \sigma_t \epsilon_t.$ $\qquad$ ▷ Improve editing quality
11: $\qquad\quad i_{\text{out}} \leftarrow x_0$
12: $\qquad\quad$ Update the parameters of Attribute Encoder $E_A$ as Equation 6
13: **return** $i_{\text{out}}$

---

We depict the attribute editing process for the attribute "glasses" facilitated by ZIP in Figure 4. The accompanying noise maps are showcased. As the temporal step $t$ progresses, the cumulative noise across 25 steps is visualized within the noise image. Each noise image is derived from a linear extraction throughout the entire generation process. Concurrently, the respective generated images are displayed. Up until $t = 600$, no $\Delta h$ is incorporated into the generation, thereby retaining the original image's style. After reaching the $t = 300$ mark, an additional point to note is that, in order to enhance the quality of the generation process, $\Delta h$ is still not introduced.

Figure 4 effectively demonstrates that pixels are more *concentrated* on attribute features. This implies the reference image's efficacy in generating visual components associated with the target attribute. Moreover, in our ZIP process, attribute generation becomes feasible following the insertion of $\Delta h$.

## 5 EXPERIMENTS

Within this section, we present empirical evidence to substantiate the effectiveness of our Zero-shot Inversion Process (ZIP) in conducting semantic latent editing across diverse attributes and datasets. Section 5.3 delves into the outcomes achieved across various datasets. Additionally, the subsequent section, Section 5.2, features comparative experiments that underscore ZIP's superior capabilities in semantic editing. This is observed across both in-domain and out-of-domain attributes.

**Baselines.** To facilitate comparison, we implement the ILVR (Choi et al., 2021) and Asyrp (Kwon et al., 2022), as benchmarks against ZIP. ILVR operates as an image-guided approach to semantic synthesis. In the forthcoming experiments, ILVR's parameters include a low-pass filter scale of $N = 64$ and a time step of $t = 100$. Asyrp, on the other hand, ascertains that diffusion models

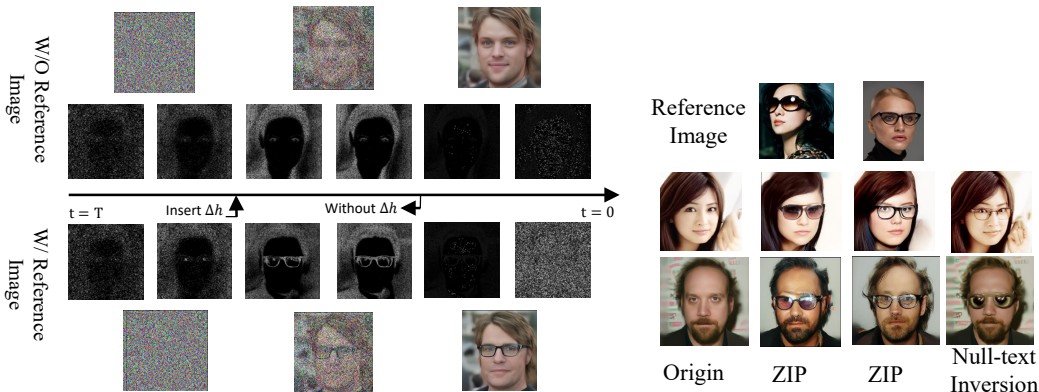

Figure 4: The visualization of noises in ZIP at different time steps $t$. The image is edited for the attribute of "glasses." The top half is the editing process without the reference image while the bottom has the reference image. Pixels are more *concentrated* on the features of the attribute, which implies that the reference image effectively works on generating visual items of the target attribute.

Figure 5: **Consistent Editing:** After specifying an image as reference image, ZIP can make a consistent and controllable editing to the original image. However, other methods, like Null-text Inversion, cannot generate specific style of the attribute.

inherently possess a semantic latent space, rendering it a state-of-the-art text-guided approach. For our Asyrp implementation, we adhere to the default parameters stipulated in the official codebase.

**Implement Details.**    All methods, including ZIP, are subjected to training on the CelebA-HQ (Karras et al., 2018), LSUN-church, and LSUN-bedroom datasets (Yu et al., 2015). In the case of ZIP, the Edit Generator draws upon DDPM++ (Song et al., 2020b), and the employed model is sourced from the official pre-trained checkpoint, thus bypassing any training specific to our experiments. Our Visual Generator is realized through Unidiffuser (Bao et al., 2023), while the Text Encoder is represented by the CLIP model (Radford et al., 2021), both using the official checkpoints. Of note, only the last five layers of the Attribute Encoder are subject to training within the ZIP framework. Further details can be explored in the supplementary materials provided.

**Evaluation.**    To assess the proficiency of image generation and editing, prior research has introduced numerous evaluation metrics. In this study, we opt to employ the Inception Score (ISC) (Salimans et al., 2016) and the Fréchet Inception Distance (FID) (Szegedy et al., 2016) as indicators of the image generation quality. Furthermore, the CLIP Score (Radford et al., 2021) is leveraged to gauge the alignment between edited images and their intended semantic targets.

## 5.1    CONSISTENCE OF ZIP

It is difficult to control the style of attributes based on text-image model, such as Prompt2Prompt (Hertz et al., 2022) and Null-Text Inversion (Mokady et al., 2022). As shown in Figure 5, with a specific reference image, ZIP generates the same styles of glasses for different images. However, Null-Text Inversion can only generate the glasses, which cannot be controlled.

## 5.2    GENERALIZATION OF ZIP

In our evaluation, ZIP is assessed from two perspectives of generalization. Firstly, we scrutinize its performance in manipulating attributes that are inherently present within the datasets. For instance, in CelebA-HQ, numerous images feature individuals with smiling expressions, and the attribute of "smiling" is explicitly labeled in the datasets. For the purposes of this evaluation, we refer to these attributes as "In-domain Attributes". Conversely, we examine attributes that are not explicitly represented in the datasets, such as "Add glasses". These are referred to as "Out-of-domain Attributes".

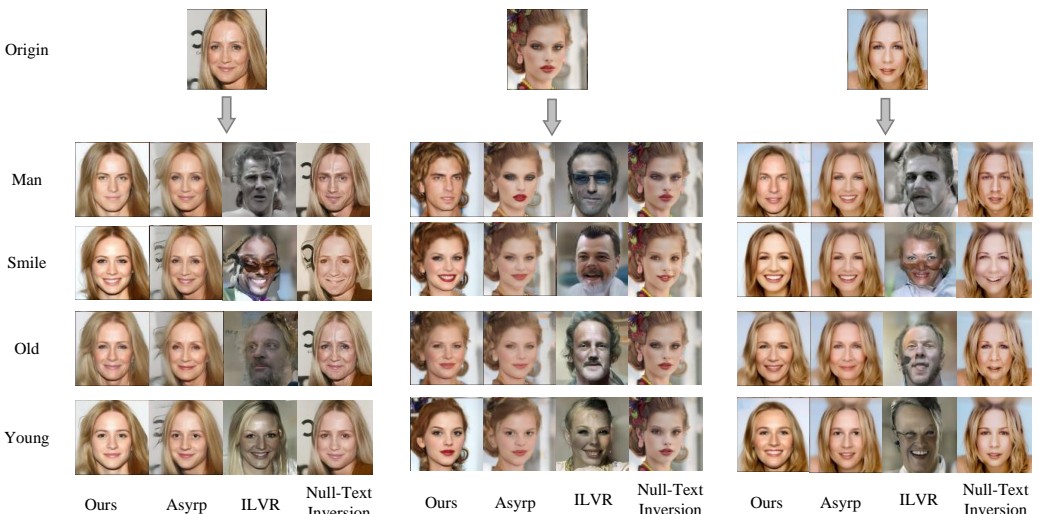

Figure 6: **Editing results for in-domain attributes**. The same attributes are modified by ZIP, Asyrp and ILVR.

Table 1: In-domain Attributes Modification

| Method | Man | | | Old | | | Smiling | | | Young | | |
|---|---|---|---|---|---|---|---|---|---|---|---|---|
| | ISC | FID | CLIP | ISC | FID | CLIP | ISC | FID | CLIP | ISC | FID | CLIP |
| ILVR (Choi et al., 2021) | 2.623 | 150.2 | 22.24 | 2.361 | 145.1 | **23.26** | 2.557 | 223.7 | 25.59 | 2.582 | 147.9 | 22.70 |
| Asyrp (Kwon et al., 2022) | 1.808 | 77.65 | 19.15 | 1.833 | 77.82 | 22.73 | 1.760 | 73.93 | 26.31 | 1.705 | 80.03 | **26.11** |
| Null-text Inversion (Mokady et al., 2022) | 2.219 | 52.23 | 23.02 | 2.075 | 50.65 | 22.46 | 2.014 | 44.25 | 25.98 | 2.089 | 40.52 | 24.42 |
| Ours | 1.592 | 87.31 | **22.38** | 1.778 | 85.96 | 22.79 | 1.573 | 85.36 | **27.01** | 1.624 | 85.28 | 25.07 |

Figure 6 presents the outcomes of our method concerning in-domain attributes, juxtaposed with results from ILVR and Asyrp. Our approach successfully alters specific attributes while keeping other attributes constant. This includes modifying expressions of individuals. In the context of text-guided approaches like Asyrp, attributes that don't necessitate alterations in visual features are handled adeptly. For example, attributes like "smiling" which entail facial feature deformations, can be effectively edited by Asyrp. However, altering attributes like gender demands replacing female features with male features, a feat that cannot be accomplished solely through text-guided methods. Meanwhile, for image-guided approaches such as ILVR, the relationship between the target attribute and the reference image can be less defined. Consequently, ILVR is prone to producing distorted or inaccurately manipulated images, as depicted in Figure 6. ZIP's strength lies in its capacity to synergize textual prompts with reference images to yield high-quality edits across a diverse array of attributes. The outcomes displayed in Table 1 corroborate this, demonstrating that ZIP achieves superior editing outcomes for attributes, without compromising on quality, when compared to alternative methods.

Figure 7 portrays the outcomes pertaining to out-of-domain attributes, such as the addition of glasses and alteration of makeup. ZIP consistently achieves the highest CLIP scores across all attributes, a fact corroborated by the data presented in Table 2 and 3. As observed with in-domain attributes, Asyrp's performance is impeded by its inability to access additional visual features. Conversely, ZIP demonstrates its competence in generating novel attributes by leveraging the capabilities of the Visual Generator and Text Encoder. This facilitates the generalization of both visual and semantic information, rendering ZIP highly proficient in addressing out-of-domain attributes.

## 5.3 MORE RESULTS

Figure 8 illustrates the visual outcomes attained through our methodology across diverse datasets. ZIP demonstrates its capacity to synthesize attributes present within the training datasets, such as age and gender in CelebA-HQ. Additionally, it can manipulate attributes based on the semantics provided by the text prompt, exemplified by "gothic" in LSUN-church. This versatility in semantic synthesis showcases ZIP's ability to accomplish varied tasks solely through training with distinct text prompts.

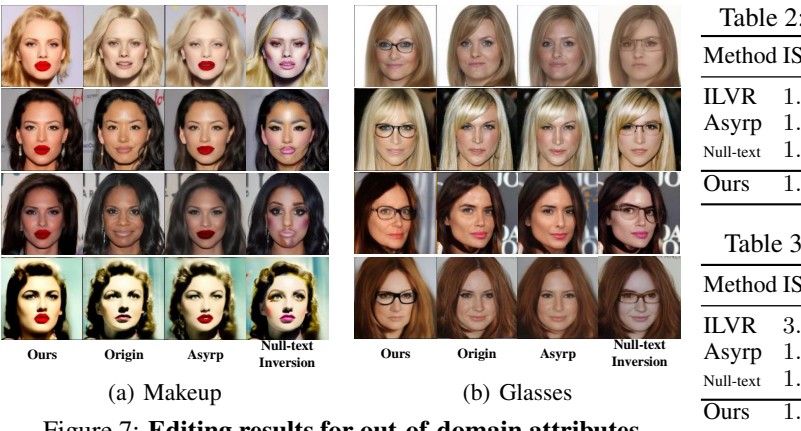

| (a) Makeup | (b) Glasses |

Figure 7: **Editing results for out-of-domain attributes.**

Table 2: **Makeup Editing**

| Method | ISC | FID | CLIP |
|---|---|---|---|
| ILVR | 1.940 | 144.4 | 24.10 |
| Asyrp | 1.755 | 77.49 | 23.86 |
| Null-text | 1.976 | 69.64 | **26.46** |
| Ours | 1.467 | 108.0 | 25.12 |

Table 3: **Glasses Editing**

| Method | ISC | FID | CLIP |
|---|---|---|---|
| ILVR | 3.060 | 165.6 | 24.52 |
| Asyrp | 1.390 | 147.5 | 29.55 |
| Null-text | 1.98 | 47.55 | 26.17 |
| Ours | 1.418 | 154.6 | **30.07** |

Figure 8: **Editing results of ZIP on various datasets.** We conduct experiments on CelebA-HQ, LSUN-church and LSUN-bedroom.

## 6 CONCLUSION

This paper introduces ZIP, a novel approach for the manipulation of real-world images using natural language. ZIP achieves this by infusing a blend of generated visual reference and textual guidance into the semantic latent space of a *frozen* diffusion model. By bridging the gap between visual patterns and textual semantics, ZIP is capable of effectively altering attributes, irrespective of whether they are in-domain or out-of-domain. In the future, our research will delve into enhancing the accuracy of attribute acquisition from reference images. This involves refining methods for specifying attributes with similar visual features, thus further improving the manipulation process.

**Limitation.** Indeed, while ZIP excels in producing realistic edits for real images, it's important to acknowledge that it faces a limitation in the absence of a target mask, similar to the capability demonstrated by Imagen Editor (Wang et al., 2022). Imagen Editor's ability to use an editing mask as input allows it to achieve intensive and precise editing of the target attributes. On the other hand, ZIP's operation without a clearly defined mask might lead to inadvertent alignment of visual features and text prompts, potentially resulting in incorrect or unintended modifications. This points to an avenue for improvement in the future, as addressing this limitation could further enhance ZIP's editing accuracy.

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
