# OpenReview forum: "Zero-shot Inversion Process for Image Attribute Editing with Diffusion Models"
_ICLR.cc/2024/Conference — ICLR 2024 Conference Withdrawn Submission_

### Official Review · Reviewer_VELF · 2023-10-28

**Soundness:** 3 good
**Presentation:** 2 fair
**Contribution:** 2 fair
**Rating:** 3
**Confidence:** 4

**Summary:**

The authors introduce a modification to the recent image-editing technique, Asyrp, aimed at enhancing its performance. In the original Asyrp method, intermediate features extracted by diffusion models are manipulated to produce images with specific attributes. These manipulations are guided by a small neural network that predicts the necessary shifts based on the original features. The proposed modification involves changing the input of this predictor from features to an image generated using a text prompt that outlines the desired attribute. This change allows the model to receive more comprehensive visual information about the attribute, resulting in improved editing performance. The experimental results, conducted across various datasets, validate the efficacy of this proposed approach.

**Strengths:**

- The main idea, which is to change the input of the attribute encoder from features to an image containing the desired attribute, is simple and intuitively reasonable. It would ease the optimization of the attribute encoder for providing accurate feature shifts to generate editted images, because the input itself has visual information of the desired attribute.

- In the experiments, the proposed method outperforms the original Asyrp in some cases, for example wearing glasses shown in Figure 7.

**Weaknesses:**

- The novelty of the proposed method is marginal. The major part of the proposed method seems identical to Asyrp.

- The effectiveness of the proposed method is not clear in quantitative evaluations. From Table 1, 2, and 3, we cannot conclude that the proposed method achieves better performance than the other methods.

- Several related studies on image editing with diffusion models are missing in reference.
  - [R1] "Imagic: Text-Based Real Image Editing with Diffusion Models," CVPR 2023.
  - [R2] "UniTune: Text-Driven Image Editing by Fine Tuning a Diffusion Model on a Single Image," SIGGRAPH 2023.
  - [R3] "Diffusion Visual Counterfactual Explanations," NeurIPS 2022.
  - [R4] "Zero-shot Image-to-Image Translation," SIGGRAPH 2023.

- The name of the proposed method is confusing, because there is no inversion process in the proposed method.

**Questions:**

Please see weaknesses.

---

### Official Review · Reviewer_nLmB · 2023-10-30

**Soundness:** 3 good
**Presentation:** 2 fair
**Contribution:** 2 fair
**Rating:** 3
**Confidence:** 5

**Summary:**

This work delves into the challenges and advancements in image editing using denoising diffusion models (DDMs). Existing methods either use image-guided techniques, which provide visual reference but lack control over semantic coherence, or text-guided methods, which ensure faithfulness to text guidance but may compromise visual quality. To address these issues, the authors introduce the Zero-shot Inversion Process (ZIP), a framework that combines both visual reference and text guidance. ZIP uses a small neural network to encode feature attribute to latent space and demonstrates effectiveness in both in-domain and out-of-domain attribute manipulation on real images. The paper provides experiments on several benchmark datasets, showing that ZIP produces images of equivalent quality while ensuring realistic editing effects.

**Strengths:**

- ZIP offers an approach by fusing both image-guided and text-guided methods, aiming to harness the strengths of both.
- The framework only requires a tiny neural network, making it computationally efficient.

**Weaknesses:**

- The integration of text guidance and image guidance in a zero-shot setting might introduce complexities in real-world applications.
- The paper only shows effectiveness on some specilized datasets, which limits the generalization ability of this method.
- It's hard to claim "in-domain" and "out-of-domain" for those attributes, since all of these attributes are existing in the datasets, the only difference is whether having explicit labels.
- The paper does not delve deeply into potential failures or edge cases of the ZIP framework.

**Questions:**

- The noise maps and insert/without $\Delta h$ in Fig 4 are confusing. How do you get noise map? Does both rows have these inserting and removing $\Delta h$ process?
- It seems the editing is attribute-wised. Do you need to train the neural network every time when applying a new attribute or every editing?
- Have you tested your method on models trained with larger datasets, such as Stable Diffusion? Does it still work?

---

### Official Review · Reviewer_pQWL · 2023-10-31

**Soundness:** 1 poor
**Presentation:** 3 good
**Contribution:** 1 poor
**Rating:** 3
**Confidence:** 5

**Summary:**

In the paper, the authors present a novel approach known as ZIP (Zero-shot Inversion Process) to effectively tackle the prevalent issue of semantic incoherence in prior methods. Employing a trainable model that maps attributes to vectors, which are then added to the DDPM model, they strive for improved visual quality while maintaining fidelity to the text. The conducted experiments, including comparisons with previous inversion techniques across diverse attributes like makeup editing and glasses editing, thoroughly assess the model's performance.

**Strengths:**

- The presentation and clarity of the content are satisfactory. The information is effectively communicated, and there are no significant issues with how it is presented.

- Figure 4 provides an interesting visualization that offers insights into the addition of the h-vector to the model.

**Weaknesses:**

- The paper lacks technical novelty as it relies on ASYRP (https://openreview.net/forum?id=pd1P2eUBVfq). This foundation alone, in my opinion, does not meet the standards for publication in ICLR. The methodology section closely mirrors that of the ASYRP paper, encompassing the training of the h-generator module and the sampling process. The asserted 'Zero Shot Inversion' process appears to be synonymous with the asymmetric reversal process outlined in the ASYRP paper, albeit with the addition of an attribute encoder.

- The experimentation is confined to DDPM, with results excluding recent foundational models such as Stable Diffusion.

- The number of attributes (man, old, smiling, young) utilized in the in-domain evaluation is insufficient. Additionally, only 2 domains (human faces and LSUN dataset) are used.

- I failed to identify any practical advantages of this approach over previous methods concerning time, diversity, and generalizability. Given the inadequacy of the experiments, I remain unconvinced.

- Table 1 illustrates no superiority of the ZIP approach over previous methods in terms of results.

**Questions:**

- I am curious if you have applied this approach to latent diffusion models on various datasets. Further elaboration on this aspect would enhance the comprehensiveness of your study.

---

### Official Review · Reviewer_L9gz · 2023-11-06

**Soundness:** 2 fair
**Presentation:** 2 fair
**Contribution:** 1 poor
**Rating:** 3
**Confidence:** 4

**Summary:**

In this paper, authors propose a zero-shot inverse process (ZIP) that can inject information from a visual reference as well as a text prompt for image editing tasks. The authors claim that their method offers more realistic and coherent image editing capabilities compared to text-to-image or image-guided methods.

**Strengths:**

1. The ZIP method shows consistent editing capabilities based on the reference image.
2. Zero-shot method

**Weaknesses:**

1. In general, the paper is not well written. Most of the technical ideas are borrowed from other works and do not offer any new insights into the problem. To point out few issues -
a.  "Compared to state-of-the-art methods, ZIP produces images of equivalent quality" Does this sentence mean their method does not offer any advantages in terms of quality?
b.  "making it challenging to confirm the exact appearance of the added glasses in advance" It is hard to comprehend what this sentence is trying to convey.
c. "Thus, our method is zero-shot and avoids the bias of manual selection" What is the bias in the context of image editing?

2. In the text authors explain Figure 2 and mention Visual Generator. I don't see any network named as a "visual generator" in Figure 2.

**Questions:**

I do not have any questions.